# CAUSAL DISCOVERY FROM CONDITIONALLY STATION­ARY TIME-SERIES

## ABSTRACT

Causal discovery, i.e., inferring underlying cause-effect relationships from obser­vations of a scene or system, is an inherent mechanism in human cognition, but has been shown to be highly challenging to automate. The majority of approaches in the literature aiming for this task consider constrained scenarios with fully ob­served variables or data from stationary time-series.

In this work we aim for causal discovery in a more general class of scenarios, scenes with non-stationary behavior over time. For our purposes we here regard a scene as a composition objects interacting with each other over time. Non­stationarity is modeled as stationarity conditioned on an underlying variable, a state, which can be of varying dimension, more or less hidden given observations of the scene, and also depend more or less directly on these observations.

We propose a probabilistic deep learning approach called State-Dependent Causal Inference (SDCI) for causal discovery in such conditionally stationary time-series data. Results in two different synthetic scenarios show that this method is able to recover the underlying causal dependencies with high accuracy even in cases with hidden states.

## 1 INTRODUCTION

The ability of deep learning approaches to discover and reason about causal relationships in data has become a prominent direction of work over the recent years (Yi et al., 2020; Girdhar & Ramanan, 2020; Sauer & Geiger, 2021). Despite the recent success of deep learning methods in related tasks such as classification, localization, and segmentation, causal discovery and reasoning, an inherent mechanism in human cognition (Spelke & Kinzler, 2007) allowing reasoning about counterfactuals and understanding the reasons of events, still poses an considerable challenge.

Causal discovery involves uncovering the underlying logic, temporal and causal structure of the observed processes in the data. Current approaches (see Section 2) commonly address quite con­strained scenarios with a stationary behavior over time. In the present paper, we extend the current work by addressing scenarios with conditional stationarity, where the dynamics of the observed system changes with the value of underlying variables. This is the case in almost all real-world scenarios, e.g. with people who behave differently and take different decisions depending on un­derlying factors such as mood, previous experience, and the actions of other agents. We propose a method (see Section 3) for causal discovery from time-series observations of systems where the underlying causal graph changes depending on a state variable.

The causal discovery task from such conditionally stationary time-series poses different challenges depending on the observability of the underlying state variable. Four scenario classes can be seen:

1. The first class concerns a simplified version of the problem, where the state variable is observed and not dependent on other observed time-series data.

2. In the second class of scenarios, the state is not directly observed, but directly dependent on and continuously inferable from an observed variable. A real-life example is a traffic scenario where taxis (visually distinguishable by the sign on their roof) follow slightly different rules than normal cars, i.e. are allowed to drive in bus lanes.

3. A more challenging scenario class is when the state depends on earlier events, and thus is not continuously observable. A real-life example is a chain of events in a football game, where the action of one player is triggered by an earlier action by another player.

4. Finally, a large share of scenarios in the real world are governed by underlying state variables that are not fully inferable from the observations from the scenario over time. In such scenarios, the state is an unknown confounder to the observed time-series, and causal discovery from such scenarios is inherently ill-defined.

We evaluate the method (see Section 4) in two different synthetic scenarios, where we vary the complexity of system dynamics and observability of the underlying state variable covering the first three scenario classes above. Finally we conclude and discuss directions for future work (see Section 5).

## 2 RELATED WORK

Causal discovery approaches aim to identify causal relationships over a set of variables from observational data. These methods can basically be classified into three different types (Glymour et al., 2019): 1) Constraint-based; 2) Score-based; 3) Functional causal model based methods.

*Constraint-based* methods rely on conditional independence tests to recover the underlying DAG structure of the data, such as the PC algorithm (Spirtes et al., 2000), which assumes faithfulness and causal Markov condition and considers i.i.d. sampling and no latent confounders. There exists a great variety of variations of PC. One of them is the Fast Causal Inference (FCI) (Spirtes, 2001), which is able to cope with the unknown confounders and selection bias; furthermore, it can be adapted for time-series data, such as such as tsFCI (Entner & Hoyer, 2010).

*Score-based* methods define score functions of causal graph structures and then optimize score functions by performing a search to identify the underlying causal structure, such as the Greedy Equivalence Search (GES) (Chickering, 2002). Notice that searching in the graph space poses a combinatorial optimization problem. Recent approaches try to avoid this by reformulating it as a continuous optimization problem which introduces a score function $h$ for measuring the acyclicity of the graph (Zheng et al., 2018). Regarding time-series data, these methods are reformulated as learning *dynamic* Bayesian Networks (DBNs) from data (Murphy et al., 2002). Among these algorithms we recently find DYNOTEARS (Pamfil et al., 2020), which aims to simultaneously estimate instantaneous and time-lagged relationships between variables in a time-series.

*Functional causal model-based* methods represent the effect as a function of its direct causes and their independent immeasurable noise (Glymour et al., 2019). For non-temporal data, there are linear non-Gaussian acyclic models (Shimizu et al., 2006), additive noise models (Peters et al., 2014), post-nonlinear models (Zhang & Hyvärinen, 2009), etc. For temporal data, these approaches fit a dynamics model, which is often regularized in terms of its sparsity, and its form is analyzed to identify the underlying causal connections in the data. Granger causal analysis falls into this category, since we first model the dynamics and some analysis is performed to extract the latent causal structure (Granger, 1969).

Causal discovery is, in general, a challenging task and its study arises a great amount of practical issues. The problem is ill-posed when considering linearity and Gaussian disturbances, since it can be proved that the underlying causal model is not identifiable, while under proper assumptions, such as non-Gaussianity, it becomes identifiable (Shimizu et al., 2006). When considering non-linear transformations, the symmetry between observed variables disappears, allowing the identification of causal relations in the context of Gaussian disturbances (Hoyer et al., 2008). Other practical issues consist on the existence of latent confounders (Ranganath & Perotte, 2018), the presence of measurement error (Zhang et al., 2017) or considering observations with missing data (Tu et al., 2019). In order to avoid these common problems, simplifications of the problem need to be applied. In fact, the assumptions we make in this work are: (i) all the instances belonging to the causal graph are observed, (ii) we have no missing data selection bias, and (iii) no latent confounders exist.

The work most related to ours are the approaches by (Löwe et al., 2020; Li et al., 2020); we extend these by allowing the causal model of the underlying process to vary depending on a state variable. Our method can in the future be applied to a wider class of non-stationary visual scenarios where the

interacting objects are only partially and noisily observed as semantically segmented visual regions, or by tracked image keypoints (Löwe et al., 2020). This would allow addressing challenging tasks such as scene understanding, counterfactual reasoning, etc. The recent work by (Sauer & Geiger, 2021) also uses the concept of causality for a similar task, generation of counterfactual images.

## 3 STATE-DEPENDENT CAUSAL INFERENCE

In this section, we introduce our formulation to extract causal graphs from time-series data where their dynamics are altered by means of a categorical variable, referred to as their state. We refer to our method as Underline{S}tate-Underline{D}ependent Underline{C}ausal Underline{I}nference (SDCI).

### 3.1 PROBLEM FORMULATION

The input consists of a set of $N$ time-series which not only obey some dynamics that might change over time but also undergo different states along the sequence. These states are responsible for the changes in the dynamics of the sample. We observe the sequence for a total of $T$ frames and we denote the sample $\mathbf{x}$ as

$$\mathbf{x} = \{\{\mathbf{x}_i^1\}_{i=1}^N, ..., \{\mathbf{x}_i^T\}_{i=1}^N\}, \quad \mathbf{x}_i^t = \{\mathbf{p}_i^t, s_i^t\}, \tag{1}$$

where $\{s_i^{1:T}\}_{i=1}^N$ represents the hidden states and $\{\mathbf{p}_i^{1:T}\}_{i=1}^N$ are the observed quantities of interest. For simplicity, we drop the subscript when referring to all the elements in a single time-step (e.g. $\mathbf{x}^t$, $\mathbf{p}^t$, $\mathbf{s}^t$, etc). In a causal graph, the observed quantities are represented by the vertices and the edge of the causal graph represents the interaction type between vertices. We denote the amount of possible interaction types by $n_\epsilon$.

**Assumptions.** In this work, we assume that the data generation process obeys a structural causal model (SCM) (Pearl, 2009), $G^{1:T}$, and that the model satisfies the first-order Markov property. Moreover, according to the definitions of causality (Granger, 1969), we assume that edges of a causal graph cannot go back in time. The first assumption follows related works concerning samples where the generative process also follows an SCM (Löwe et al., 2020; Li et al., 2020). Although we assume the first-order Markov property, one can extend to the more general $p - th$ order Markov property in a more complex scenario.

**State-dependent causal inference.** Based on the assumptions we mainly focus on the non-stationary causal graph, which means that we can find different edge-types at different times. As for an edge between two vertices, the edge-type interaction between two vertices changes according to the state of the variable which is the source of the interaction. The main focus of our method consists on extracting a *causal summary graph* $G$ (also denoted as such by Li et al. (2020) and Löwe et al. (2020)). Previous approaches aiming for this task assume stationary time-series data and therefore, this *causal summary graph* is constant. Nonetheless, since we condition the stationarity of the samples on the states, our *causal summary graph* is expressed by means of this categorical variable. In other words, our method will extract $K$ summary graphs, one per state considered. The edge-type interaction can be then queried at each time-step $t$ as follows:

$$z_{ij}^t = G_{ij}(s_i^t) \tag{2}$$

where $z_{ij}^t \in \{0, ..., n_\epsilon - 1\}$ denotes the edge-type interaction from $i$ to $j$ at time-step $t$. Figure 1 illustrates this task. In general, this *causal summary graph* is specific to the input sample and hidden from the model. Therefore, not only we require to design a parametrizable function to infer the latent causal structure, but also to evaluate how this inference fits to the actual dynamics observed in the input sequence.

Let us denote the first step of extracting the latent causal structure

$$G(\mathbf{s}) = f_\phi(\mathbf{p}^{1:T}, \mathbf{s}^{1:T}) \tag{3}$$

where $f_\phi$ denotes a parameterizable function that receives all the observed sequence as input. The next step is to fit this extracted latent sturcture into our assumed first-order Markov dynamics.

$$(\tilde{\mathbf{p}}^{t+1}, \tilde{\mathbf{s}}^{t+1}) = f_\psi(\mathbf{x}^t, G(\mathbf{s}^t)) \tag{4}$$

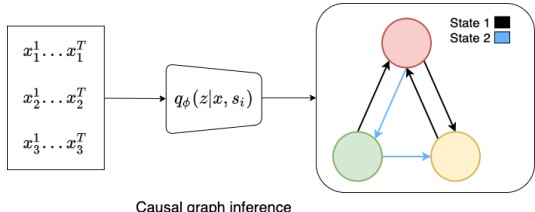

Causal graph inference

Figure 1: SDCI aims to extract a *causal summary graph* that describes the edge-type interaction for every pair of edges conditioned on the state of the source variable with respect to the interaction.

where the parameterizable function $f_\psi$ represents a one-step computation of the dynamics starting from the observed values at time-step $t$. In this expression, we have defined $f_\psi$ to predict the value of the states at the next time-step as well. However, in our experiments we will also consider settings where the states are observed at all times and modelling the dynamics is only performed with respect to the quantity $\mathbf{p}^{1:T}$. To allow for this setting, one only needs to exclude the state variable from supervision. We provide more details in the next section.

**Objective.**  Both the causal inference and dynamics modelling modules can be optimized by minimizing some objective defined for the parameterizable functions $f_\phi$ and $f_\psi$.

$$\min_{\phi,\psi} \sum_{t=1}^{T-1} \mathcal{L}(\mathbf{x}^{t+1}, f_\psi(\mathbf{x}^t, G(\mathbf{s}^t))) + R(G(\cdot)), \quad G(\mathbf{s}^t) = f_\phi(\mathbf{x}) \tag{5}$$

where $R(\cdot)$ is a regularizer on the extracted graph structure, which can be applied to enforce a preferred interaction type.

## 3.2  Implementation

Since the interactions that are considered in the scene are complex, we introduce uncertainty in our approach. Therefore, we are interested in computing the probability distribution over the edge type given two instances and the state of the source instance, $p(z_{ij}|s_i, \mathbf{p}_i, \mathbf{p}_j)$. This implementation gathers inspiration from related works concerning this type of tasks (Li et al., 2020; Löwe et al., 2020; Kipf et al., 2018).

The structure of the implementation is very similar to a VAE (Kingma & Welling, 2014). In this case, the latent space is represented by the set of edges $z_{ij}$ which are discrete variables. The latent edges are conditioned to the state of the variable which is the source of the interaction, $s_i$. We allow for a total of $K$ states. Let $z_{ij}$ be an edge from instance $i$ to instance $j$. This value represents the type of causal interaction that there is between $i$ and $j$ and will depend on the state of $i$. $z_{ij}^t \in \{0, ..., n_\epsilon - 1\}$, where $n_\epsilon$ is the amount of inter-object interactions considered in our setting. $z_{ij}^t = 0$ means that component $i$ is not influenced by component $j$ at time $t$. Any other value models a different type of interaction. Since we condition the edge type on the state variable, our objective is more close to the one defined in CVAE (Sohn et al., 2015).

$$\mathcal{L} = \mathbb{E}_{q_\phi(\mathbf{z}|\mathbf{s},\mathbf{x})}\big[\log p_\psi(\mathbf{x}|\mathbf{z})\big] + KL\big(q_\phi(\mathbf{z}|\mathbf{s},\mathbf{x})||p(\mathbf{z}|\mathbf{s})\big) \tag{6}$$

where $p(\mathbf{z}|\mathbf{s})$ is a prior defined over the edge types conditioned on each state, which acts as a regularizer over the inferred edge-type distribution. We use $\mathbf{z}$ to denote all the edges represented in the latent space. In our settings, we set this prior to enforce a uniform distribution no matter what the state is. We might find applications where this prior could be leveraged to encourage sparsity in the extracted graph structure.

**Encoder.**  Following related approaches (Löwe et al., 2020), we use graph neural networks as the encoder. The model receives all the information from the sample available $\mathbf{x}^{1:T}$. We concatenate $\mathbf{p}^{1:T}$ with a one-hot representation of the state variable $\mathbf{s}^{1:T}$. We aim to extract an embedding that represents the causal interaction conditioned on the state for every pair of elements present in the sample:

$$\phi_{ij} = f_\phi(\mathbf{x}^{1:T})_{ij} \tag{7}$$

where $\phi_{ij} \in \mathbb{R}^{n_\epsilon \times K}$ denotes the embedding for every pair $i \rightarrow j$. The approximate posterior distribution conditioned on the state variable $q_\phi(z_{ij}|k, \mathbf{x})$ is calculated as follows:

$$q_\phi(z_{ij}|k, \mathbf{x}) = \Theta(\phi_{ijk}/\tau) \tag{8}$$

where $k$ is the state to which the edge-type distribution of $i \rightarrow j$ is conditioned and $\Theta$ denotes a softmax activation with temperature $\tau$. We use $\phi_{ijk} \in \mathbb{R}^{n_\epsilon}$ to the note the embedding for $i \rightarrow j$ at state $k$. Since this formulation yields a discrete latent space, we relax the resulting categorical distribution using the Gumble-softmax (Maddison et al., 2017) technique to enable back-propagation.

Once we have inferred the approximate posterior distribution over the edge-types, we can sample the interactions at every time-step $t$ conditioning on $s^t$:

$$z_{ij}^t \sim q_\phi(z_{ij}|s_i^t, \mathbf{x}) \tag{9}$$

Notice that this requires sampling at each time-step $t$, which might pose great computational expenses for large sequences. In practice, we sample right after determining the distribution of the edge-types. Then, the interactions are queried for each $i \rightarrow j$ depending on $s_i^t$:

$$w_{ijk} \sim q_\phi(z_{ij}|k, \mathbf{x}), \quad z_{ij}^t = w_{ijk'}, \quad k' = s_i^t \tag{10}$$

where we have stored the sampled values using $w_{ijk} \in \mathbb{R}^{n_\epsilon}$, which denotes the edge-type using a one-hot representation. In our experiments, we refer to SDCI-Static or SDCI-Temporal for implementations based on MLPs or CNNs respectively. For more details, see the supplementary material.

**Decoder.** Let us describe the decoder $p_\psi(\mathbf{x}|\mathbf{z})$, which models the dynamics of the generative process. Consider the dynamics modelling for element $j$. At each time $t$, we first query the edge-types depending on the states of the source variable $i$ from the sampled quantities $w_{ijk}$ (see equation 10).

$$z_{ij}^t = w_{ijk}, \quad k = s_i^t \tag{11}$$

The information along the predicted edge-type interactions is then retrieved as follows

$$\mathbf{g}_{ij}^t = \sum_{e>0} \mathbf{1}_{(z_{ij}^t = e)} f_e(\mathbf{x}_i^t, \mathbf{x}_j^t) \tag{12}$$

$\{f_e\}_{e=1}^{n_\epsilon - 1}$ is a family of parametrizable functions, one defined for each edge type excluding the no-edge interaction.

The inter-object interactions are finally integrated to model the dynamics of each variable. The updates of the variable $\mathbf{p}_j^{t+1}$

$$\tilde{\mathbf{p}}_j^{t+1} = \mathbf{p}_j^t + f_p\Big(\sum_{i \neq j} \mathbf{g}_{ij}^t, \mathbf{x}_j^t\Big) \tag{13}$$

are predicted by means of $f_p$, where it provides information about the update between the actual and the next time-step. The supervision of the decoder considering $\mathbf{p}^{1:T}$ uses the negative log-likelihood of the following probability distribution

$$p_\psi(\mathbf{p}_j^{t+1}|\mathbf{x}^t, \mathbf{z}^t) = \mathcal{N}(\tilde{\mathbf{p}}_j^{t+1}, \sigma^2 \mathbb{I}) \tag{14}$$

where $\sigma$ is the variance of the resulting Gaussian distribution.

We have previously introduced the idea of considering the state variable $\mathbf{s}^{1:T}$ as being independent from the dynamics of the sample, i.e. $\mathbf{p}^{1:T}$. If this happens, our method will have information about the states of the samples at all times and consequently, no supervision upon $\mathbf{s}^{1:T}$ will be required. However, we might find practical to consider the setting where the state is in fact dependent on $\mathbf{p}^{1:T}$, and therefore our method will require to model its behavior and perform supervision. In this case, we present a straightforward approach to compute the estimation of the next state:

$$\tilde{s}_j^{t+1} = f_s\Big(\sum_{i \neq j} \mathbf{g}_{ij}^t, \mathbf{x}_i^t\Big) \tag{15}$$

and the corresponding categorical distribution can be computed as follows

$$p_\psi(s_j^{t+1}|\mathbf{x}^t, \mathbf{z}^t) = \Theta(\tilde{s}_j^{t+1}) \tag{16}$$

where $\Theta$ is a softmax activation.

In summary, the probability distribution of the decoder can be expressed as follows:

$$p_\psi(\mathbf{x}|\mathbf{z}) = \prod_{t=1}^{T-1} p_\psi(\mathbf{x}^{t+1}|\mathbf{x}^t, \mathbf{z}^t), \quad p_\psi(\mathbf{x}^{t+1}|\mathbf{x}^t, \mathbf{z}^t) = p_\psi(\mathbf{p}^{t+1}|\mathbf{x}^t, \mathbf{z}^t)p_\psi(\mathbf{s}^{t+1}|\mathbf{x}^t, \mathbf{z}^t) \quad (17)$$

Notice that with this formulation, $\mathbf{p}$ and $\mathbf{s}$ can be expressed as separate factors. We find this result useful, since in practice we calculate the log-likelihood of both probability distributions separately and balance the resulting loss term

$$\mathcal{L} = \mathbb{E}_{q_\phi(\mathbf{z}|\mathbf{s},\mathbf{x})}\big[\log p_\psi(\mathbf{p}|\mathbf{z})\big] + \lambda\mathbb{E}_{q_\phi(\mathbf{z}|\mathbf{s},\mathbf{x})}\big[\log p_\psi(\mathbf{s}|\mathbf{z})\big] + KL\big(q_\phi(\mathbf{z}|\mathbf{s},\mathbf{x})||p(\mathbf{z}|\mathbf{s})\big) \quad (18)$$

where we use $\lambda$ to balance the contribution of both terms to the optimization process.

**Hidden state regime.** When considering the state as a variable hidden from the input data, we can still sample from the estimated posterior distribution as follows:

$$z_{ij}^t \sim \sum_{k=1}^K q_\phi(z_{ij}|k, \mathbf{x})p(k|\mathbf{x}), \quad s_i^t \sim p(k|\mathbf{x}) \quad (19)$$

where $p(k|\mathbf{x})$ is the probability distribution which models the states at each time-step. In this setting, we modify Eq. 16 and condition solely on the object dynamic variables.

$$p_\psi(s_i^t|\mathbf{x}^t) = \Theta(\tilde{s}_i^t/\gamma), \quad \tilde{s}_i^t = f_{s'}(\mathbf{x}_i^t) \quad (20)$$

$\gamma < 1$ is a temperature factor which increases the confidence of the model predictions. Since we should be sampling from Eq. 19 and we consider $p(k|\mathbf{x}) = p_\psi(k|\mathbf{x}^t)$, in practice we have

$$z_{ij}^t = \sum_{k=1}^K w_{ijk} \cdot p_\psi(k|\mathbf{x}^t) \quad (21)$$

## 4 EXPERIMENTS

We now present the experiments that have been performed to evaluate the method and compare it to the baseline method, amortized causal discovery (ACD) (Löwe et al., 2020). All models have been implemented using Pytorch (Paszke et al., 2019) and all training and test processes have been carried out on NVIDIA RTX 2080Ti GPUs.

### 4.1 EXPERIMENTS ON LINEAR DATA

We start with a simple scenario with linear message passing operations between a number of different time-series, where each time-step $t$ in series $i$ is a one-dimensional continuous variable $\mathbf{p}_i^t \in \mathbb{R}$.

The variables are connected by edges of $n_\epsilon$ different types. Each edge-type represents a different causal effect that one variable can perform to another. This is denoted by $\{\beta_k\}_{k=1}^{n_\epsilon - 1}$. $\beta_0$ represents no connection, i.e., no causal interaction between a pair of variables. At each time-step $t$, each element in the sample performs the following operation to update $\mathbf{p}_i$:

$$\mathbf{p}_i^{t+1} = \alpha\mathbf{p}_i^t + \sum_{i \neq j}^N \beta_k\mathbf{p}_j^t \quad (22)$$

where $\alpha \in \mathbb{R}$ controls the self-connection, and $\beta_k \in \mathbb{R}$ represents the edge-type connection between $j$ and $i$, of type $k$.

We carry out experiments in this scenario to compare the results obtained by the two proposed architectures and the baseline method ACD Löwe et al. (2020). We are interested in observing whether our formulation is capable of recovering the underlying parameters of the linear message passing mechanism and identify the causal interactions in each sample in different types of scenarios (see Section 1). In this case, we experiment with observed states that are independent from the observations, and hidden states (first and third scenario classes). Unless noted otherwise, we set $K = 2$ states.

Table 1: Edge-type accuracy (in %), reconstruction MSE, and distance to world parameters using linear data for 3 variables, 2 states, and 2 edge-types.

| METHOD | EDGE ACCURACY | | RECONST. MSE | | DIST. TO WORLD |
|---|---|---|---|---|---|
| | TRAIN | TEST | TRAIN | TEST | |
| ACD (Löwe et al., 2020) - FIXED DECODER | $66.35 \pm 0.12$ | $66.02 \pm 0.29$ | $0.49 \pm 9.01 \cdot 10^{-3}$ | $0.49 \pm 1.89 \cdot 10^{-2}$ | - |
| ACD (Löwe et al., 2020) | $66.90 \pm 0.13$ | $66.44 \pm 0.29$ | $0.47 \pm 8.60 \cdot 10^{-3}$ | $0.47 \pm 1.98 \cdot 10^{-2}$ | $5.62 \cdot 10^{-3}$ |
| SDCI - STATIC - FIXED DEC. | $90.69 \pm 0.10$ | $90.43 \pm 0.23$ | $2.23 \cdot 10^{-2} \pm 1.31 \cdot 10^{-3}$ | $2.64 \cdot 10^{-2} \pm 4.55 \cdot 10^{-3}$ | - |
| SDCI - STATIC | $93.84 \pm 0.09$ | $93.84 \pm 0.19$ | $1.34 \cdot 10^{-2} \pm 1.13 \cdot 10^{-3}$ | $1.57 \cdot 10^{-2} \pm 4.03 \cdot 10^{-3}$ | $6.60 \cdot 10^{-6}$ |
| SDCI - TEMPORAL - FIXED DEC. | $82.97 \pm 0.13$ | $82.79 \pm 0.28$ | $7.03 \cdot 10^{-2} \pm 1.62 \cdot 10^{-3}$ | $7.43 \cdot 10^{-2} \pm 4.79 \cdot 10^{-3}$ | - |
| SDCI - TEMPORAL | $49.92 \pm 0.13$ | $49.97 \pm 0.28$ | $0.86 \pm 1.61 \cdot 10^{-2}$ | $0.84 \pm 3.29 \cdot 10^{-2}$ | $2.18 \cdot 10^{-2}$ |

### 4.1.1 DATA GENERATION

The procedure for generating this dataset is as follows. First, we set the edge-type interactions. In our experiments we set $\alpha = 1$ and since we experiment with 2 edge-types, we set $\beta_1 = 0.05$. To generate each sample, we need to sample the initial values of the continuous variable for each element $\mathbf{p}_i^0$ and the underlying causal structure dependent on the state, $\mathcal{G}(s)$. At each time-step, it suffices to query the edge-type $k$ for each pair of variables and apply the corresponding causal effect $\beta_k$ following Equation 22. The edge-type is $k = \mathcal{G}(s_j^t)_{ji}$, where $\mathcal{G}(s)_{ji}$ denotes the causal effect from $j$ to $i$, which has been defined at the beginning of the sequence. For all our experiments with this dataset, we simulate $N = 3$ variables for $T = 40$ time-steps. We decide to keep $N$ and the interaction values ($\beta_k$) low because the generated samples are unstable, which imply that the data is not restricted within a certain range and could be problematic for higher values of $N$ and $T$.

When considering hidden states, the state is updated as follows: $s_i^t = \mathbf{1}_{(\mathbf{p}_i^t < 0)}$.

### 4.1.2 TRAINING SPECIFICATIONS

All the models participating in the experiments of this section have been trained following the same training scheme, including ACD (Löwe et al., 2020).

**Customized decoder.** One of our objectives is to recover the underlying world parameters $\beta_k$. Thus, we implement a decoder which imitates the message passing operation presented in Equation 22, which allows us to initialize the decoder using the underlying world parameters and analyse the performance of the encoder as a separate entity from the whole model.

**Model parameters.** Following Kipf et al. (2018), the models have been trained using ADAM optimizer (Kingma & Ba, 2015). The learning rate of the encoder is $5 \cdot 10^{-4}$, the learning rate of the decoder is $1 \cdot 10^{-3}$, and both are decayed by a factor of 0.5 every 200 epochs. We train for 1000 epochs using a batch size of 128. We use teacher forcing every 10 time-steps during training. This implies that the decoder receives the ground-truth as input every 10 time-steps, otherwise it uses its previous output. The temperature $\tau$ is set to 0.5 and the variance of the Gaussian distribution of the decoder for $\mathbf{p}$ is $\sigma^2 = 5 \cdot 10^{-5}$. When considering the setting where we make the state dependent on the dynamics of the objects, we set $\lambda = 10^3$. For hidden states, we set the temperature $\gamma = 0.1$.

### 4.1.3 RESULTS

We firstly consider two edge-types and only one state to search for suitable learning rates for the encoder and the decoder. The edge-type accuracy of our method considering SDCI-Static for this setting is 94.88% on training data and 94.87% on test data. Since SDCI-Static is equivalent to ACD (Löwe et al., 2020) in this setting, we use this value as a reference for the following experiments.

We now proceed to comparing the performance between the two architecture choices SCDI-Static and SCDI-Temporal with the baseline ACD, and evaluate the effect of explicitly modeling the underlying state. Furthermore, we compare the two proposed methods SCDI-Static and SCDI-Temporal with a a variant where the decoder is fixed and uses the ground-truth parameters, $\beta_k$. We will in the following compare all these models considering two edge-types.

Table 1 shows the edge-type identification accuracy, the distance to the world parameters, and the reconstruction MSE for three variables, two states, and two edge-types (no-edge and $\beta_1$). As we can see, our method SDCI-Static successfully performs the task of identifying the state-dependent causal interactions. Furthermore, it recovers the underlying parameters of the generative process

with great accuracy. On the other hand, our SDCI-Temporal is not able to identify properly the causal interactions nor recover the underlying parameters with enough accuracy. However, when fixing the decoder using the true underlying parameters, it achieves decent performance. Regarding our predecessor (ACD), we observe that its performance is considerably lower than our proposed architectures. One should note that this formulation can still recover the underlying world parameters, but less accurately compared to our formulation with SDCI-Static.

We also experimented with SDCI-Static when considering the state variable hidden from the input data. The results show an edge-type identification accuracy of $92.25\%$ and $91.96\%$ in training and test data respectively. The distance to the world parameters is $1.06 \cdot 10^{-4}$. To assess the effectiveness of the model to identify different behaviors (states) when considering this setting, we can inspect the estimation of the latent state probability distribution. The state decoder accuracy of the latent state function is $99.10\%$ and $98.97\%$ in training and test data respectively. We can observe that our formulation allows considering hidden state variables. SDCI-Static is able to successfully identify the causal interactions between the elements and successfully decomposes the different behaviors into the actual underlying states of the generative process.

## 4.2 EXPERIMENTS ON SPRING DATA

In the second experiment setting, we evaluate our methods on data similar to that used in recent work (Kipf et al., 2018; Löwe et al., 2020), consisting of particles (or small balls) connected by springs with directed impact - meaning that e.g. particle $i$ could affect particle $j$ with a force through a connecting spring, but leaving particle $i$ unaffected by this spring force.

The following experiments focus solely on evaluating the performance of our SDCI-Static and comparing it with its predecessor (ACD) under the three first scenario classes introduced in the Introduction. The data generation process follows the description of Kipf et al. (2018). The only difference is the addition of the state variable to the generative process, which affects the edge-type at each time-step as in the previous case. For more details, see the supplementary material. As in the first experimental setting, the experiments regarding this data always consider 2 edge-type connections (presence/absence of directed spring) and a sequence length of $T = 80$.

### 4.2.1 TRAINING SPECIFICATIONS

We use the same training scheme for all the models present in the experiments of this section. In this case, the configuration is identical to the one used by Kipf et al. (2018). Thus, the experiments have been trained using ADAM optimizer (Kingma & Ba, 2015). The learning rate of both the encoder and decoder is $5 \cdot 10^{-4}$ and decayed by a factor of 0.5 every 200 epochs. We train for 500 epochs using a batch size of 128. We also use teacher forcing every 10 time-steps during training. The temperature $\tau$ is set to 0.5 and the variance of the Gaussian distribution of the decoder for $\mathbf{p}$ is $\sigma^2 = 5 \cdot 10^{-5}$. When considering the setting where we make the state dependent on the dynamics of the objects, we set $\lambda = 10^3$. For hidden states, we set $\gamma = 0.05$.

### 4.2.2 RESULTS

Let us consider the first scenario class, where the state is known and independent from the observations. The state transitions incrementally into the next one every 10 time-steps, and alternative settings with up to 8 states are explored. Table 2 shows the corresponding results, where for each for state (from 1 to 8) a new dataset is generated and used to train the SDCI-Static method. We observe that with one state, the original results reported by Löwe et al. (2020) are obtained since this case corresponds to stationary time-series. As the number of states increase, the accuracy in edge-type identification decreases and the reconstruction MSE becomes larger. However, our SDCI-Static is able to maintain promising results achieving $74.87\%$ accuracy in edge-type identification when having as much as 8 states.

We now proceed to the scenario class number 3, where the state of a particle transitions when it collides with the wall of the box where it is contained. For simplicity, we only consider two states that transition alternatively on wall collision. Our proposed method SDCI-Static achieves an edge-type identification accuracy of $85.13\%$ on training data and $79.21\%$ on test data. The reconstrucion MSE is $1.32 \cdot 10^{-4}$ and $1.38 \cdot 10^{-3}$ on training and test data respectively. Since now our decoder

Table 2: Edge-type accuracy (in %) and reconst. MSE using spring data with different states for SDCI-Static.

| NUM. STATES | EDGE ACCURACY | | RECONST. MSE | |
|---|---|---|---|---|
| | TRAIN | TEST | TRAIN | TEST |
| 1 | $99.70 \pm 0.02$ | $99.67 \pm 0.13$ | $7.88 \cdot 10^{-5} \pm 5.38 \cdot 10^{-6}$ | $7.88 \cdot 10^{-5} \pm 4.64 \cdot 10^{-4}$ |
| 2 | $98.80 \pm 0.02$ | $97.11 \pm 0.08$ | $8.00 \cdot 10^{-4} \pm 1.53 \cdot 10^{-5}$ | $4.02 \cdot 10^{-2} \pm 1.96 \cdot 10^{-4}$ |
| 3 | $98.00 \pm 0.03$ | $95.79 \pm 0.09$ | $2.50 \cdot 10^{-3} \pm 3.26 \cdot 10^{-5}$ | $2.33 \cdot 10^{-2} \pm 1.64 \cdot 10^{-4}$ |
| 5 | $89.88 \pm 0.04$ | $80.34 \pm 0.10$ | $5.36 \cdot 10^{-3} \pm 2.98 \cdot 10^{-5}$ | $6.57 \cdot 10^{-2} \pm 3.23 \cdot 10^{-4}$ |
| 8 | $79.37 \pm 0.03$ | $74.87 \pm 0.08$ | $9.59 \cdot 10^{-3} \pm 3.50 \cdot 10^{-5}$ | $3.02 \cdot 10^{-2} \pm 1.63 \cdot 10^{-4}$ |

requires modelling the conditions for state transition, we can evaluate its performance in identifying state changes. The state accuracy of the decoder is $99.73\%$ on training data and $98.53\%$ on test data, which shows that our method is able to detect the effect that the event of a particle colliding with the wall will have on the whole sample. If we consider ACD, it achieves an edge-type accuracy of $69.87\%$ on training data and $68.61\%$ on test data. The reconstruction MSE is $4.45 \cdot 10^{-4}$ and $1.46 \cdot 10^{-3}$ on training and test data respectively, and the state accuracy of the decoder is $99.13\%$ on training data and $98.21\%$ on test data. As we can see, the limitation of ACD of considering only stationary time-series data limits the performance of the causal graph inference task. However, the results for the reconstruction MSE are comparable to the ones obtained with our formulation (SDCI-Static), which indicates that the method still makes decent predictions although it fails in identifying the edge-type interactions.

Finally we address a scenario of class 2 as listed in the Introduction. In this scenario, the underlying state of a particle changes depending its location in the box; thus, the state is not directly observable but indeed directly dependent on an observable variable, the position of the particle. Our proposed method SDCI-Static shows an edge-type identification accuracy of $85.35\%$ and $80.82\%$ on training and test data respectively. The reconstruction MSE is $1.44 \cdot 10^{-4}$ and $1.19 \cdot 10^{-3}$ on training and test data respectively. Notice that in this case, we also find accuracy levels that are similar to the second and third layout. If we perform inspection on the estimation of the latent state variable distribution, we observe a state decoder accuracy of $98.96\%$ in training data and $99.41\%$ in test data. As we can see, our formulation allows to learn the dynamics of the state without any supervision. Regarding ACD, it achieves an edge-type accuracy of $71.06\%$ in training data and $69.60\%$ in test data. The reconstruction MSE is $4.11 \cdot 10^{-4}$ and $1.31 \cdot 10^{-3}$ in training and test data respectively. As before, ACD is restricted since it considers a stationary generative process. However, the behavior of the sequences in this data regime is non-stationary as the state variable is hidden. Our SDCI-Static successfully decomposes the non-stationary dynamics into the true underlying conditional stationary ones and is able to identify the causal links between particles successfully enough.

## 5 CONCLUSIONS

In this work we propose a method, SDCI, for performing causal discovery in scenes with multiple objects interacting over time, and where the dynamics depend on the value of underlying states. SDCI allows recovering the underlying causal structure of non-stationary time-series data by conditioning its stationarity on categorical state variables. Furthermore, we provide a deep probabilistic implementation of this method and perform an empirical study on two synthetic scenarios. Our results show the effectiveness of our formulation in modelling conditional time-series data, in comparison to the state-of-the-art approaches that consider stationary time-series data.

### 5.1 FUTURE WORK

This work can be regarded as a preliminary but necessary contribution towards causal discovery in video. In order to approach this goal, we propose to add a visual front-end as a pre-processing step of the SDGI module. This front-end would operate on the video and output probabilistic segmentation of video into object hypotheses, pose extraction of humans in the scene, detection and recognition of known object classes, etc.

This automatic extraction of causal abstractions of natural scenes would enable taking steps towards challenging tasks such as high-level scene understanding or counterfactual reasoning.

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
