# OpenReview forum: "Causal discovery from conditionally stationary time-series "
_ICLR.cc/2022/Conference — ICLR 2022 Submitted_

### Official Review · Reviewer_ukPw · 2021-11-02

**Correctness:** 2
**Technical Novelty And Significance:** 2
**Empirical Novelty And Significance:** 2
**Recommendation:** 3
**Confidence:** 4

**Main Review:**

Overall, the considered setting is interesting, but the paper organization and writing need to be improved. For example, in some parts, it seems that the authors are considering instantaneous causal relations, while in some parts, it seems the authors are considering time-delayed causal relations. It is not clear to the readers. Also, I am not sure about the state variable, whether it is observed or not, because it is not consistent throughout the paper; for example, in Section 3.2 which discusses the implementations, the state variable seems to be observed.

Another concern is about theoretical identifiability. When claiming causality, it is essential to show the theoretical properties, e.g., identifiability of the causal graph. However, the authors did not give any theoretical results in the paper.

Some related references are missing, which consider causal discovery in a similar setting:
1.	B. Saeed, S. Panigrahi, and C. Uhler. Causal Structure Discovery from Distributions Arising from Mixtures of DAGs. ICML, 2020.
2.	B. Huang, K. Zhang, J. Zhang, J. Ramsey, R. Sanchez-Romero, C. Glymour, and B. Schölkopf. Causal Discovery from Heterogeneous/Nonstationary Data. JMLR, 2020.


**Summary Of The Paper:**

This paper considers causal discovery from nonstationary time series, where the nonstationarity is due to some unobserved state factors. To handle this setting, the authors propose an approach based on probabilistic deep learning.

**Summary Of The Review:**

Interesting setting, but essential theoretical results are missing, and the paper organization and writing needs improvement (some parts are inconsistent throughout the paper)

---

### Official Review · Reviewer_quAM · 2021-11-03

**Correctness:** 3
**Technical Novelty And Significance:** 3
**Empirical Novelty And Significance:** 2
**Recommendation:** 5
**Confidence:** 4

**Main Review:**

Strengths:
- The problem domain of inferring causal structure from time-series in a non-stationary context is a very important one, and the most realistic setting that one can expect.
- The proposed method depends on existing methods, meaning that is a reasonable improvement that is likely to be reproducible.The idea of conditioning on a categorical state variable is a nice and logical one

Weaknesses:
- I believe that the biggest and very important weakness is the inherent limitations of the model regarding data complexity, as the experiments also show. While a general framework is provided, I doubt if the model can truly account for non-stationary temporal processes to infer causal graph structure. That is a very hard problem, which is already hard in non-temporal processes with interventions. For instance, is a K-dimensional categorical variable (for K states) sufficient for encoding a latent state from complex observational data x? I suppose that the model could be extended for other types of variables or distributions, but is unclear if that would really work in practice (and in that sense if the proposed model is the right direction).
- It seems that currently the only latent variables encoding information about the generation process are the edge variables and the state variables, assuming they are not observed in the respective setting. If they are both categorical, do they have the necessary capacity to model complex temporal patterns observed in x?
- Extending on the previous points, the experiments are limited to synthetic and rather low complexity cases, although as I said above, this is a hard problem and likely harder settings would be not possible likely. The first experiment is on linear, one-dimensional observational data with K=2 states, while the second experiment is also low dimensional in the observation space (I think 2 dimensions) and up to 8 state variables. What would happen if the observation space is much higher dimensional, say a few dozens, and inflicted with noise?
- In the experiments, I think it is assumed that the number of states is known in advance and equal to K. Is the case of unknown or even varying K explored? Or what if K is set to be much higher than the actual number of states, just to cover all bases? Would that lead to a collapse, of certain useless dimensions? Or would it cause the model to mistakenly assign causal graphs to all states regardless.
- As an encoder, graph neural networks are used. I assume that these have a fixed, fully-connected topology, but it would describe
 more clearly the precise connectivity. I also assume the connectivity is fixed across experiments. Are there any vanishing/exploding gradients observed by increase sequence length?
- I am not sure about the validity of some statements. In the beginning of the related work it is said that ‘Causal discovery approaches aim … from observational data’. I am not sure if this is precise enough, as typically interventional data is also used, at least when not in a temporal setting.


**Summary Of The Paper:**

This paper proposes a new method for discovering the causal graph from time-series data when the time-series are generated by a non-stationary process. The method relies on previous work from Lowe et al, 2020 and proposes to condition the causal summary graph driving the (causal) edge generation between variables by a categorical state variable. The method is defined within a variational inference framework, where edges are state-dependent latent variables, based on which one can generate/reconstruct future observations and/or state variables. Results are shared on two synthetic datasets with promising performances.

**Summary Of The Review:**

I like the approach, but I am not sure if it is ready enough for publication. I thus initially recommend borderline reject.

---

### Official Review · Reviewer_15XZ · 2021-11-03

**Correctness:** 2
**Technical Novelty And Significance:** 2
**Empirical Novelty And Significance:** 2
**Recommendation:** 5
**Confidence:** 3

**Main Review:**

While the paper seems to provide an interesting and novel method, it also seems to extend the work of [Lowe et al 2020] to conditionally stationary time-series in a relatively straightforward fashion.

One of the issues I have with this paper is that it fails to address if the model is indeed identifiably causal or just a graph that fits well the data. In [Lowe et al 2020] the graph is shown to be causal under  the assumptions of first-order Markovianity, no latent confounders, no instantaneous relations and assuming the optimization procedure finds the global optimum. In this case, the authors should provide similar results, taking especially into account the fact that there might be latent confounders (e.g. the s_i might play this role if they are related with other s_j). Moreover, it isn't clear to me if the latent states can be uniquely reconstructed in a way that would render the time series conditionally stationary.

Another issue I have is that it doesn't adequately cover the related work, for example there are other methods that model non-stationarity in a similar way, e.g. https://arxiv.org/abs/1903.01672. This also extends to the evaluation seems to be focused almost exclusively on [Lowe et al 2020] (which unsurprisingly doesn't work if the data is non-stationary).

**Summary Of The Paper:**

The paper presents a causal discovery method for time-series that generalizes [Lowe et al 2020] to the conditionally stationary case. In particular, the method assumes there is a latent variable $s_{i}^t$ for each timeseries $i$ that influences the graph and edge types at each timestep, such that conditioning on this variable the time-series is stationary.

**Summary Of The Review:**

In summary the paper is promising, but not ready yet. A better embedding in the existing research, a few theoretical results (mostly about making explicit under which assumptions the graphs learnt are actually causal) and a more thorough evaluation would improve it a lot.

---

### Official Review · Reviewer_BArP · 2021-11-06

**Correctness:** 3
**Technical Novelty And Significance:** 2
**Empirical Novelty And Significance:** 2
**Recommendation:** 3
**Confidence:** 3

**Main Review:**

This work is based on state-dependent causal inference for time series (TS). The goal is to extract a causal summary graph $G$ (Fig. 1) where the edge type can change based on state. The proposed method is based on conditional VAE. It considers an auxiliary variable s (states of all the edges in one TS) which is conditioned on by the prior.

Strengths:
1. Very detailed experimental setup is given for each experiment, which helps reproducibility.
2. It considers two evaluation tasks at one time -- causal graph extraction and TS reconstruction (not forecasting).

weakness:
1. The presentation really really needs improvements.
2. It only compares with one baseline ACD, only in linear data. In the spring data they did not compare the proposed method with anything but just show a sensitivity analysis on the number of states.


# details:
1. Without a figure illustrating the model structure, it is very hard to interpret what variables like $k$ stand for and why do we need to have such variables.
2. It is very hard to understand what is the input and output of the function $G$.
3. It really needs clarification on all the notations. For example, what is the difference between $G$ and $\mathbf{g}_{ij}^t$?

# questions:
1. What is a state?
2. Which variable is the source variable in Fig. 1?
3. Why is $s_j^t$ not an input for $G_{ij}$ in eq.(2)? If this is the case, how do you guarantee $G_{ji}$ and $G_{ij}$ are consistent with each other. E.g., you cannot have $i$ causes $j$ and $j$ also causes $i$ at the same time.
4. Why is $G$ a function if it is a graph?
5. Would the dimension of $z$ be quite high since it represents all the edges among the variables.





**Summary Of The Paper:**

This paper aims to solve the problem of causal summary graph extraction and time series reconstruction at the same time. They propose a conditional VAE based model. The model is conditioned on state variables $s$, which makes it different from a normal VAE. Experiments on two datasets show the method outperforms ACD on linear data and its performance drops as number of states increase with spring data.


**Summary Of The Review:**


Strengths:
1. Very detailed experimental setup is given for each experiment, which helps reproducibility.
2. It considers two evaluation tasks at one time -- causal graph extraction and TS reconstruction (not forecasting).

weakness:
1. The presentation really really needs improvements.
2. It only compares with one baseline ACD, only in linear data. In the spring data they did not compare the proposed method with anything but just show a sensitivity analysis on the number of states.

---

### Decision · Program_Chairs · 2022-01-20

**Decision:**

Reject

**Comment:**

This paper extends Lowe et al. 2020 to discover causal relations from nonstationary time series by assuming conditionally stationarity of the times series. Based on the assumption, a deep learning method based on VAE is proposed to learn the causal relations from data.
The paper is well-motivated and well-organized.

However, there are some concerns from the reviewers. 1) The presentation needs significant improvement, e.g., clarification of the notations.  2) The identifiability of the causal graph is not given. It is unclear under what conditions the proposed method can discover the true causal graph. 3) The capacity of the discrete states might not be able to handle complex real situations. 4) The experiments are limited to synthetic and low complexity cases. This further weakens the significance of the proposed method given that there are also no theoretical guarantees of the proposed method. 5) Discussions about some important relevant works are missing.

Overall, the paper studies an interesting problem. However, given the above concerns, the novelty and significance of the paper will degenerate. Both theoretical and empirical analysis of the proposed method need further improvement. Addressing the concerns needs a significant amount of work. Thus, I do not recommend acceptance of this paper.